# NGLSFusion: Non-Use GPU Lightweight Indoor Semantic SLAM

Le Wan [1,2], Lin Jiang [1,3,*], Bo Tang [2,3], Yunfei Li [1,2], Bin Lei [1,3] and Honghai Liu [4]

1   Key Education Laboratory of Ministry of Metallurgical Equipment and Control,
    Wuhan University of Science and Technology, Wuhan 430081, China;
    wl1446220918@gmail.com (L.W.); liyunfei20180607@163.com (Y.L.);
    leibin@wust.edu.cn (B.L.)
2   Hubei Key Laboratory of Mechanical Transmission and Manufacturing Engineering,
    Wuhan University of Science and Technology, Wuhan 430081, China; tang1017@163.com
3   Institute of Robotics and Intelligent Systems, Wuhan University of Science and Technology,
    Wuhan 430081, China
4   School of Mechanical and Electrical Engineering and Automation, Harbin Institute of Technology,
    Shenzhen 518055, China; honghai.liu@hit.edu.cn
*   Correspondence: jianglin76@wust.edu.cn; Tel.: +86-15927625084

**Abstract:** Perception of the indoor environment is the basis of mobile robot localization, navigation, and path planning, and it is particularly important to construct semantic maps in real time using minimal resources. The existing methods are too dependent on the graphics processing unit (GPU) for acquiring semantic information about the indoor environment, and cannot build the semantic map in real time on the central processing unit (CPU). To address the above problems, this paper proposes a non-use GPU for lightweight indoor semantic map construction algorithm, named NGLSFusion. In the VO method, ORB features are used for the initialization of the first frame, new keyframes are created by optical flow method, and feature points are extracted by direct method, which speeds up the tracking speed. In the semantic map construction method, a pretrained model of the lightweight network LinkNet is optimized to provide semantic information in real time on devices with limited computing power, and a semantic point cloud is fused using OctoMap and Voxblox. Experimental results show that the algorithm in this paper ensures the accuracy of camera pose while speeding up the tracking speed, and obtains a reconstructed semantic map with complete structure without using GPU.

**Keywords:** semantic map; non-use GPU; SLAM; RGBD-based VO; indoor environment

## 1. Introduction

As intelligent mobile robotics teams continue to evolve and robot-oriented technologies become more diverse, measuring their ability to perceive the geometry of objects inside a scene, attach accurate semantic labels, and reconstruct the recognized objects with semantic information in a map is a test of the robot's ability to properly understand its environment. When a robot comes to work in an unfamiliar environment, the ability to build an accurate map helps it identify where obstacles are and locate itself in the environment, and the semantic information it obtains provides the robot with the ability to understand its surroundings and carry out commands given by humans, such as "bring the food there" and "clean the trash to the left of the refrigerator". There are many excellent 3D reconstruction algorithms available today, such as Multi-View Stereo [1], Structure from Motion [2], SLAM [3], etc. Meanwhile, there are also breakthroughs in semantic segmentation algorithms based on deep learning, such as [4–7], but there is not much intersection between these two fields. In recent years, more attention has been focused on combining 3D reconstruction algorithms with deep learning algorithms and building semantic SLAM systems, and so far, these two research fields have been revitalized by combining with

each other. The more popular ones at this stage are dynamic semantic SLAM [8,9] and 3D semantic SLAM [10–12]. However, the combination of the two fields also brings some new problems. The 3D reconstruction algorithms rely on CPU, various algorithms based on deep learning rely on GPU, and the combination of the two will make the whole system work at high intensity, while mobile robots are mostly used as commercial service-oriented robots, considering their commercial cost performance, so high performance processors will not be configured on the robots. For this problem, this paper proposes a lightweight indoor semantic map construction algorithm non-use GPU named NGLSFusion. In the past few years, visual SLAM has been widely used, mainly because its framework is very stable and mature, and the visual-based images store rich texture information, in which some excellent algorithms have also achieved good results, such as ORB-SLAM2 [13], ORB-SLAM3 [14], VINS-Mono [15], etc. Compared to most current visual odometry (VO) and visual–inertial odometry (VIO) or 3D semantic reconstruction algorithms, NGLSFusion combines both, and the algorithm is capable of accurate localization and real-time map building, relying only on the CPU on mobile robots with low equipment performance. The VO part of the algorithm uses an excellent open-source visual SLAM, ORB-SLAM2 [13]. In this paper, an RGBD camera is used because it can provide accurate depth and scale information to improve the robustness of the system. In the algorithm, the non-keyframes coming in from the front end are combined using ORB features [16], optical flow method, and direct method to jointly obtain the keyframes and extract the feature points on the keyframes, and then PnP-Ransac is used to optimize the camera pose for the keyframes and feature points. LinkNet [17], a lightweight network using encoding and decoding connections, is used to identify indoor environmental objects to obtain object semantic segmentation results, which fits well with our mobile robot platform due to its real-time segmentation and fast operation. Since the algorithm in this paper is designed to run in real time, without GPU support, the segmentation effect will be much reduced, so we use Intel's neural network gas pedal OpenVINO to optimize the pretrained model and speed up the model inference, so as to achieve real-time recurrent inference without relying on GPU. Indoor semantic map construction is performed using OctoMap [18] (for robot navigation) and Voxblox [19] (for building more detailed textured maps).

Therefore, the algorithm given in this paper consists of a total of three modules: a VO module based on RGBD, a lightweight semantic segmentation module with real-time loops, and a 3D reconstruction module that fuses semantic information to generate maps. In conclusion, the algorithm in this paper is modular, and the modules can be freely combined to meet the requirements according to the work required when using the robot.

Considering the portability integrity on robotic platforms, NGLSFusion supports both online builds using the Robot Operating System (ROS) [20] and test validation using offline datasets. The main contributions made by the algorithms in this paper are summarized as follows:

(1) A VO module with improved ORB-SLAM2 for fast extraction of image feature points and excellent robustness. The module firstly initializes the system with ORB features for the first frame of the front input, secondly tracks the feature points on subsequent frames using optical flow method, eliminates mismatching using forward and reverse optical flow and creates new key frames, then extracts the feature points on key frames using direct method and calculates their descriptors. Finally, the camera pose is optimized by using the PnP-Ransac algorithm for keyframes and feature points, which can speed up the extraction of feature points and optimize the camera pose.

(2) A GPU-independent optimized inference lightweight model for constructing semantic point clouds. A file containing a specific network topology and a binary file containing weights and deviations are obtained by optimizing the lightweight LinkNet model, and the model is repredicted using an inference engine that reads both files and implements parallel operations using SIMD to meet real-time performance on a CPU-only device. Two-dimensional image segmentation is performed on the objects in the indoor environment acquired by the camera, and the segmented semantic colors and

their confidence levels are compared using the optimal value comparison method to generate a three-dimensional point cloud with semantic information by combining the depth information corresponding to each pixel in the image.

(3)     A lightweight semantic SLAM system for fusing semantics based on optimized models in indoor environments. The system can run in real time on devices without GPU and build maps using globally consistent pose and semantic information to ensure robot localization accuracy and map construction, and the entire system has complete debugging and visualization tools.

The other sections of this paper are structured as follows: In Section 2, the related work is presented. In Section 3, the methods used for the three modules are described. In Section 4, the proposed modules are tested experimentally and the results are analyzed rationally. In Section 5, the experimental conclusions and future perspectives are given.

## 2. Related Work

In recent years, the fusion of geometric maps of indoor environments with semantic information to build semantic maps to link objects with semantic concepts has become a popular research area in the field of mobile robotics, but there is no SLAM system that runs in real time without using GPU and concurs with both robot pose and semantic information. Indoor semantic SLAM generally consists of two modules: a VO (or VIO) module based on visual SLAM, and a semantic point cloud fusion module.

When the mobile robot is in an unknown environment, VO then locates the current location of the robot and draws a sparse map of the environment. The basic idea is to extract and match the feature points of different frames of the same scene through a continuous frame with translation and rotation, calculate the transformation between frames, and select and save keyframes for relocalization, loop closure, and map building. There are two main methods for VO (or VIO) to extract feature points from images: feature point method and direct method. The application of the feature point method in VO (VIO) can be traced back to some SLAMs based on monocular vision [21–25]. Among them, MonoSLAM [21] and RT-SLAM [22] use extended Kalman filter (EKF) to implement SLAM. Sim [23] proposed an SLAM method based on particle filter to improve the effect of system nonlinearity to EKF by introducing a traceless Kalman filter, but it increased the computational complexity. PTAM [24] chose to use nonlinear optimization as the back-end-dominant, and introduced keyframe mechanism and multithreading mechanism to improve the efficiency of the algorithm when the mainstream visual SLAM used EKF filter to process the back-end data at that time, but it was susceptible to motion blur and camera rotation and only suitable for small scenes due to its 2D–2D image matching algorithm, and did not consider the global loop. ORB-SLAM [25] extended on this basis by using ORB features to establish short-term and medium-term data associations, establishing co-views to increase visual relationships and map construction, and using the bag-of-words database DBoW2 [26] for loop closure and relocation to achieve long-term data association. Since monoculars cannot obtain accurate scale information, and the constructed maps are inaccurate, ORB-SLAM2 [13] adds stereo camera and an RGBD camera to obtain an absolute scale. ORB-SLAM3 [14] fuses the camera and the inertial measurement unit (IMU) on this basis for joint optimization of the camera pose. In terms of feature point extraction, eSLAM [27] accelerates ORB feature point extraction on an FPGA [28] platform. GCNv2 [29] is improved on the basis of the deep learning network GCN [30] to speed up feature point matching. SuperPoint [31] proposed a fully convolutional neural network architecture that uses the self-supervised domain of homographic adaptation to train images to detect key points and descriptors. Although the theory of feature point method is mature and widely used, it is difficult to match feature points for no texture or weak texture, and the extraction of feature points consumes a lot of resources. Therefore, an optical flow method was proposed, which uses optical flow to track key points instead of calculating descriptors, and it is obviously superior to the feature point method in time consumption. When processing frames, it needs to ensure that the ambient luminance is constant and that there is continuous "small motion" between

frames, which can be subdivided into dense optical flow (such as Horn–Schunck optical flow method [32]) and sparse optical flow (such as Lucas–Kanade optical flow method [33]), depending on its working principle. Based on the optical flow method, the direct method mentioned above was thus created, and both estimate the camera motion by minimizing the photometric error, except that the direct method further optimizes the camera pose using the least squares method. LSD-SLAM [34] is the first algorithm to apply the direct method in a semi-dense monocular SLAM, which achieves real-time semi-dense reconstruction based on the relationship between map gradients and the direct method. SVO [35] is fast, but has large errors and no loopback detection. DSO [36] uses a sliding window consisting of keyframes at the back end to obtain accurate camera pose even in weakly textured scenes, running fast and with high robustness.

Three-dimensional reconstruction of the indoor environment is the foundation of path planning and navigation for mobile robots. The most direct method is to acquire the point cloud through the RGBD camera on board the robot, and as the number of point clouds saved increases, the robot itself will have insufficient memory, and this method obviously cannot meet the requirements. In recent years, there has been an increasing interest in 3D reconstruction based on RGBD cameras. KinectFusion [37], as the first 3D reconstruction algorithm, relies on depth maps alone to build 3D models without relying on RGB maps, which is based on the principle of using truncated signed distance field (TSDF) to continuously move the camera to obtain objects from different viewpoints for reconstruction, but it relies on GPU parallel computation. Kintinuous [38] adds loopback detection and loopback optimization to this, enabling 3D reconstruction of large scenes, combining iterative closest point algorithm (ICP) and direct method for estimation of camera pose, but inevitably still using GPU implementation. ElasticFusion [39] was the first to represent the map as a surfel model when most 3D reconstruction algorithms use a mesh model to fuse point clouds for reconstruction. ElasticReconstruction [40] processes the input continuous frames in modules and removes the noise of the depth map according to the frames in each module to obtain more accurate reconstruction results. InfiniTAM [41] uses a hash table to store the implicit volume representation, which greatly saves the memory consumption during reconstruction. BundleFusion [42] adopts a parallelized optimization framework, makes full use of the correspondences extracted based on sparse features and dense geometry and photometric matching, and uses bundle adjustment [43] (BA) in the backend to optimize the pose, and its reconstruction results are the best among the current static scenes. Depth information alone cannot enable a robot to locate accurately in an unfamiliar environment, so it is necessary to link the reconstructed map with the semantic information of the objects in the environment, where the semantic reconstruction uses a camera, such as SemanticFusion [44], Qi [45], DynaSLAM [46], MaskFusion [47], Co-Fusion [48], and the method of semantic reconstruction using LiDAR, such as SegMap [49], SuMa [50], SuMa++ [51], and OverlapNet [52]. The above methods rely on GPU processing regardless of whether they use vision or laser for semantic reconstruction, which limits the application in robotics. Of course, there are also some CPU-based methods, such as [53–55], among which Voxblox++ [53] requires pose estimation using sensors such as RGBD cameras and wheel speed odometers to construct a Voxblox [19] map, and it relies on the GPU. DXSLAM [54], used to extract feature points, and Kimera [55], used to obtain semantic information, are obtained offline. Therefore, these three methods cannot be called semantic SLAM in the strict sense. In summary, the NGLSFusion proposed in this paper has made some improvements in solving these problems, and it has been proven to be more effective through experiments. All the methods used are described in detail in Section 3.

## 3. Method

In this section, four aspects are presented. Firstly, the complete algorithmic framework diagram of NGLSFusion is given. Secondly, the improved VO module is introduced to guarantee its pose estimation accuracy while achieving fast feature tracking. Then, the LinkNet pretraining model is optimized to provide semantic information on CPU in real

time, the best semantic information is computed using the optimal value comparison method, and the semantic point cloud is generated by combining the depth information. Finally, a lightweight semantic map construction method is proposed, which uses OctoMap and Voxblox to fuse the semantic point clouds.

### 3.1. NGLSFusion Algorithm Framework

As shown in Figure 1, the algorithm in this paper takes RGB frames and depth frames as inputs and obtains globally consistent poses from the RGBD-based VO module, giving a fast and robust pose-estimation method. The RGB frames are processed using an optimized lightweight network model to give a method to obtain the best semantic information. Using globally consistent poses, semantic information and depth information are combined to generate local semantic maps, and local semantic maps are fused to generate global semantic maps. The algorithm contains a total of three modules, each of which is given in this section, and it is the first method to achieve real-time semantic mapping on mobile robots without relying on GPU. A detailed description of the three modules is given in the following sections.

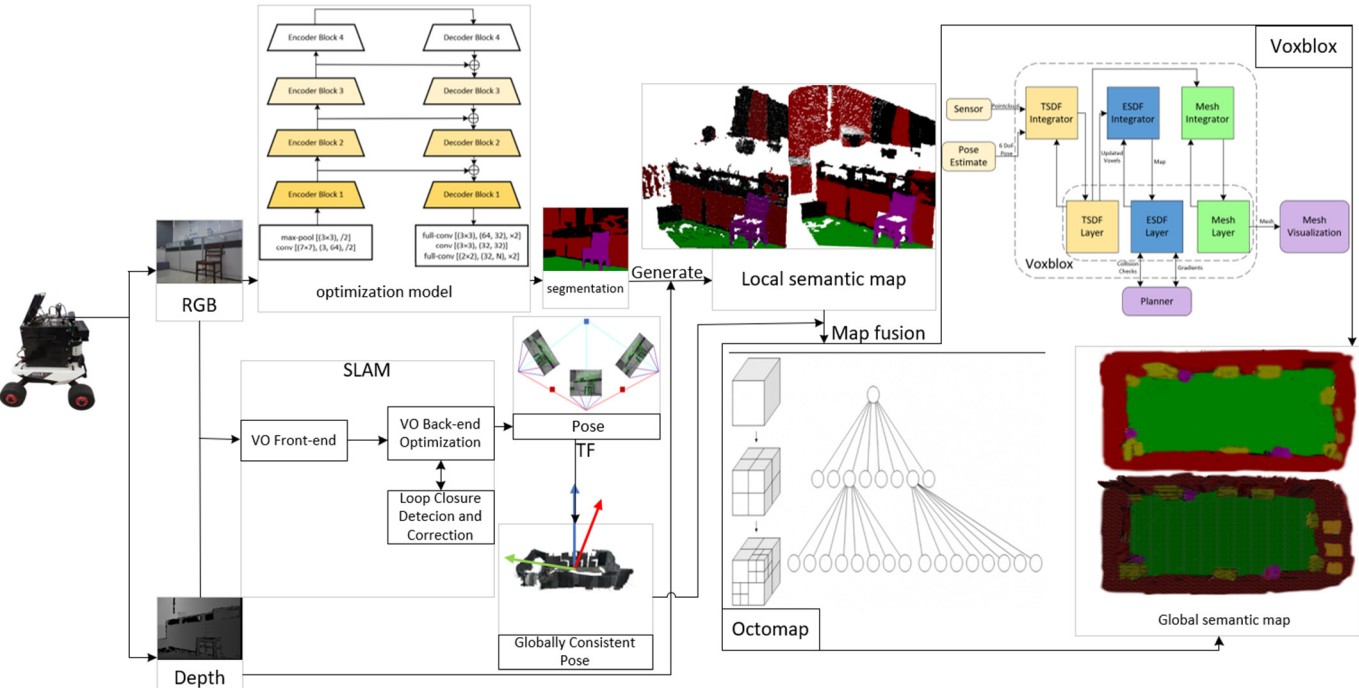

**Figure 1.** Algorithm framework of NGLSFusion.

### 3.2. Visual Odometry

The VO module used in the algorithm is improved based on ORB-SLAM2, and its algorithm framework is shown in Figure 2, where green is the input, red is the output, and yellow and blue are the judgment and processing, respectively.

The module is implemented by initializing the first frame of the input into the module using ORB feature extraction due to its rotation, illumination, and scale invariance, extracting key points using FAST [56] features, then comparing the point pairs selected in a fixed pattern around the key points to obtain a binary BRIEF descriptor [57], and then combining the obtained feature points with their depth information to generate map points using the EPNP algorithm [58] to complete the initialization of the first frame.

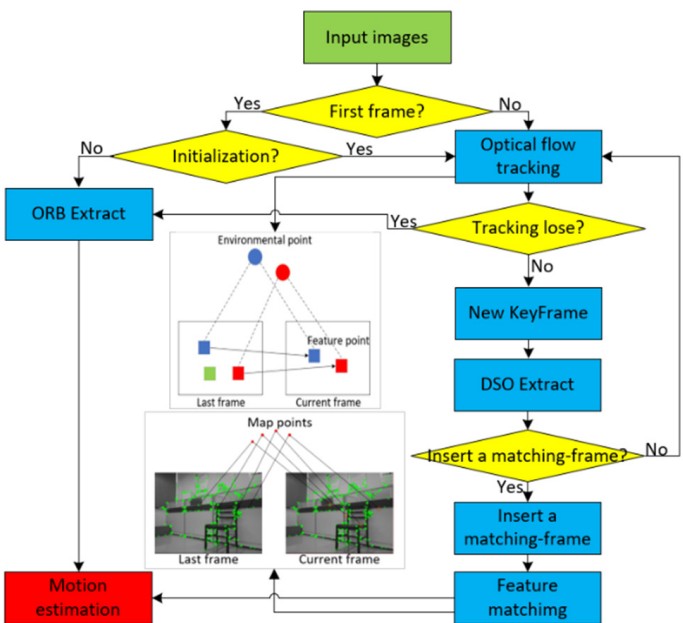

**Figure 2.** Algorithm framework of VO module.

After initialization, the Lucas–Kanade optical flow method [33] is used to track the feature points on the subsequent non-keyframes, assuming that the grayscale value of the feature point in the previous frame at moment t is denoted as $G(x, y, t)$, where $(x, y)$ denotes the coordinates of the feature point in the incoming image in the two-dimensional plane. After time $\Delta t$, the grayscale value of the same feature points in the current frame are represented by $G(x + \Delta x, y + \Delta y, t + \Delta t)$, which leads to the following equation:

$$G(x + \Delta x, y + \Delta y, t + \Delta t) = G(x, y, t) \tag{1}$$

The Taylor expansion of Equation (1):

$$G(x + \Delta x, y + \Delta y, t + \Delta t) = G(x, y, t) + \frac{\partial G}{\partial x}dx + \frac{\partial G}{\partial y}dy + \frac{\partial G}{\partial t}dt + o(dx, dy, dt) \tag{2}$$

According to Equations (1) and (2), the following relationship is obtained:

$$\frac{\partial G}{\partial x}dx + \frac{\partial G}{\partial y}dy + \frac{\partial G}{\partial t}dt + o(dx, dy, dt) = 0 \tag{3}$$

Dividing both sides of Equation (3) by d simultaneously yields:

$$\frac{\partial G}{\partial x} \cdot \frac{dx}{dt} + \frac{\partial G}{\partial y} \cdot \frac{dy}{dt} + \frac{\partial G}{\partial t} = 0 \tag{4}$$

where $\frac{\partial G}{\partial x}$ and $\frac{\partial G}{\partial y}$ denote the partial derivative of the grayscale value of the feature point to the coordinates $x$ and $y$ in the two-dimensional plane, $\frac{\partial G}{\partial t}$ denotes the partial derivative to time $t$, $\frac{dx}{dt}$ and $\frac{dy}{dt}$ denote the displacement of the feature point in the previous frame, and $o(dx, dy, dt)$ denotes the minimum of the coordinates $x$ and $y$ and time $t$, and the displacement can be used to locate the position of the same feature point in the current frame and obtain the correspondence of the feature point. If there is a large movement, the same feature points of the previous frame and the current frame are easily mismatched, so it is necessary to use the reverse optical flow to eliminate the error. As shown in Figure 3 below, the feature points 1, 2, and 3 in the previous frame are tracked by forward and reverse optical flow to determine their positions, such as 1 corresponds to 4, 2 corresponds to 5, and 3 corresponds to 6 in the forward optical flow; 4 corresponds to 7, 5 corresponds to

8, and 6 corresponds to 9 in the reverse optical flow. The error between 3 and 9 by forward and reverse tracking is already greater than 0.5 pixels, which is defined as a wrong tracking point and executed to delete it, so as to improve the optical flow tracking accuracy.

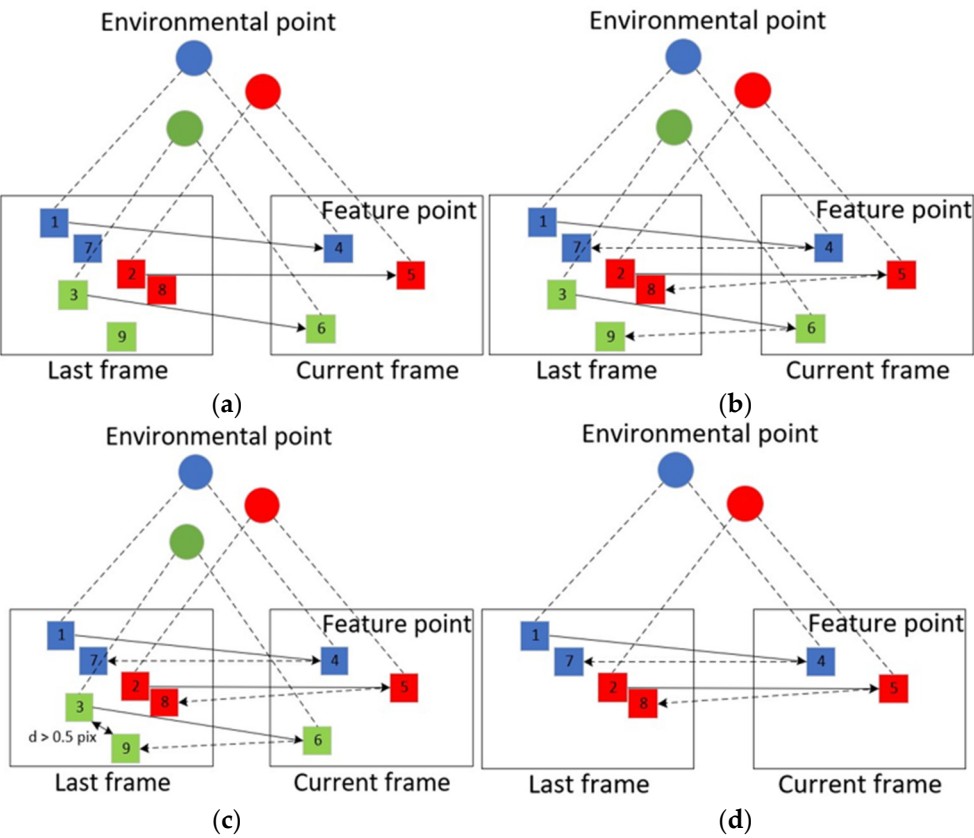

**Figure 3.** Optical flow tracking. (**a**) Forward tracking; (**b**) reverse tracking; (**c**) judgment of pixel distance; (**d**) removal of mismatched points.

New keyframes are created according to the matching relationship of feature points on the extracted images, and, finally, the feature points are extracted and their descriptors are calculated for the newly created keyframes using the direct method of DSO [36]. After obtaining enough feature points, the camera pose is calculated and optimized using the 3D–2D PnP-Ransac algorithm, assuming that there are n feature points obtained by the direct method of tracking keyframes, where the 3D points and feature points correspond as follows:

$$s_i \begin{pmatrix} u_i \\ v_i \\ 1 \end{pmatrix} = K \left( R \begin{pmatrix} X_i \\ Y_i \\ Z_i \end{pmatrix} + t \right), i = 1, 2, \ldots, n. \tag{5}$$

where $s_i$ is the depth factor corresponding to the selected sensor, $(u_i, v_i, 1)^T$ is the feature point location in the 2D image, $(X_i, Y_i, Z_i)^T$ is the 3D point in space, $K$ is the camera intrinsics matrix, and $R$ and $t$ are the camera extrinsics matrices.

Then, the pose to be optimized and the 3D point to be generated are used as the variables to be optimized for nonlinear optimization. Assuming that $T_{k,k-1}$ is the pose between adjacent frames, the 2D plane coordinates of a feature point in the $I_{k-1}$ image $(u, v)$, and the depth is $d$, the 3D coordinates $p_{k-1}$ of the feature point in the $I_{k-1}$ coordinate system are obtained by Equation (5), the 3D coordinates $p_k$ in the $I_k$ coordinate system are obtained by using the transformation matrix $T_{k,k-1}$, and the 2D plane coordinates in the $I_k$ image coordinate system $(u', v')$ are obtained by combining with the camera internal reference to complete the reprojection process. Since there is an error in the above points obtained by reprojection error, which also leads to unequal brightness values before and

after projection, this error is reduced by continuously optimizing the poses, and, finally, the preoptimized poses are obtained, which are derived as follows:

$$T_{k,k-1} = \arg\min_{T_{k,k-1}} \frac{1}{2} \sum_{i \in R} \|\delta I(T_{k,k-1}, u_i)\|^2 \tag{6}$$

Among them:

$$\delta I(T, u) = I_k(\pi(T \cdot \pi^{-1}(u, d_u))) - I_{k-1}(u) \tag{7}$$

According to the position of the feature points in the previous 2D image and their depth information, the 3D coordinate points are back-projected to the 3D space, and the 3D coordinate points are converted to the current coordinate system by using the transformation matrix; finally, they are projected back to the current frame, and the preoptimized pose $T_{k,k-1}$ is obtained by continuous optimization iterations. The keyframe pose and the map points observed on the keyframes are used as vertices; each vertex is linked with an edge, and solved by general graph optimization [59]. Finally, the loop results are obtained by comparing the similarity between the two frames using DBoW2 [26] based on the descriptors saved by computation in each keyframe. Although there are redundant calculations in creating new keyframes and extracting keyframe feature points, the optical flow method and direct method based on photometric error are much faster than the feature point method, as shown in the later experimental sections, and they have higher robustness and higher speed.

The module uses color and depth images with a resolution of $640 \times 480$ as the front-end input data, estimates the motion between adjacent frames through VO, and passes them to the back end for optimization and loop detection, which determines whether the robot "knows" the current environment and passes the acquired information to the back end for processing, so as to obtain a globally consistent pose and maps through loop closure and relocalization.

### 3.3. Semantic Segmentation

The original semantic segmentation algorithm LinkNet [17] used in this paper is based on the implementation of Pytorch [60]. The network uses the autoencoder idea to connect each encoder and decoder directly without using pooling index or full convolution to recover the stepwise convolution, removing unnecessary spatial information and greatly reducing the processing time and the number of parameters required by the network. On the basis of such a lightweight real-time segmentation algorithm, this paper uses Intel's Open Visual Inference & Neural Network Optimization (OpenVINO) tool, as it provides a complete deep learning inference toolkit (DLDT), which can be used for a variety of trained models are deployed, but models trained by Pytorch deployment are not yet supported by it, so it needs to be preprocessed in advance to convert the pretrained model format to Open Neural Network Exchange (ONNX). The pretrained model is first optimized using Model Optimizer to obtain a file containing the specific network topology and a binary model file containing the weights and biases. The Inference Engine is then used to read the above two files and repredict the model. Finally, parallel operation is achieved using single instruction multiple data (SIMD) on the CPU to meet the real time, and the corresponding optimized semantic segmentation algorithm framework is shown in Figure 4.

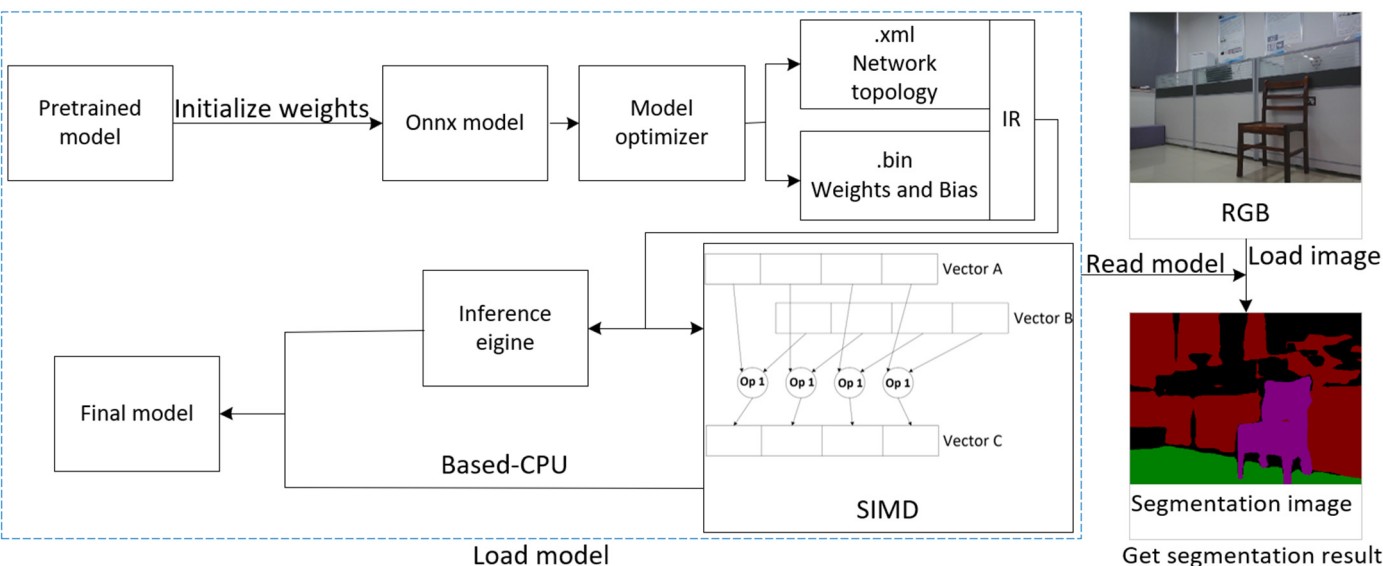

**Figure 4.** Algorithm framework of optimized semantic segmentation.

Each frame of the input color map is predicted using an optimized model that returns semantic information for each pixel in real time, where the semantic information includes the semantic color (the color given in advance to the object category) and the confidence. An optimal value comparison method is introduced to preprocess the semantic information for each pixel, and the algorithmic structure of this method is given below (Algorithm 1).

---

**Algorithm 1:** Optimal value comparison

---

1:     **function** Optimal Value Comparison ($semantic_1, semantic_2$)
2:     **if**     $semantic_1.color == semantic_2.color$
3:         $semantic_{optimal}.color = semantic_1.color$
4:             $semantic_{optimal}.conf = \frac{semantic_1.conf + semantic_2.conf}{2}$
5:     **else**
6:         **if**     $semantic_1.conf > semantic_2.conf$
7:             $semantic_{optimal}.color = semantic_1.color$
8:             $semantic_{optimal}.conf = semantic_1.conf$
9:                 $semantic_{optimal}.conf = semantic_{optimal}.conf \times 0.95$
10:        **else**
11:            $semantic_{optimal}.color = semantic_2.color$
12:            $semantic_{optimal}.conf = semantic_2.conf$
13:                $semantic_{optimal}.conf = semantic_{optimal}.conf \times 0.95$
14:    **return** $semantic_{optimal}$

---

### 3.4. Map Construction

In this paper, we fully consider the semantic map building module and propose to build two forms of maps simultaneously. One is the OctoMap map, which is mainly a occupancy grid, and the other is the Voxblox map, which is based on TSDF and can better construct the environmental texture information. The former can be used to accomplish autonomous indoor navigation tasks, while the latter provides maps with more texture information. While constructing two types of maps, the map building process is divided into building local and global maps. The local point cloud is first generated based on the combination of pixels on the color map with their corresponding depth information, and the point cloud is transformed into a local occupancy grid and a local TSDF at a certain resolution using undersampling processing. Then, each keyframe in Section 3.3 is used to obtain semantic information using semantic segmentation and combine it with its depth information to generate a local 3D semantic point cloud, which is then fused

with the local occupancy grid map and local TSDF map to generate a local semantic map, respectively. Finally, combining the current frame pose of the VO module with globally consistent pose and semantic information, the local semantic map is fused and spliced to generate the global semantic map, and the whole process is implemented on the CPU. The corresponding semantic map construction algorithm framework is shown in Figure 5.

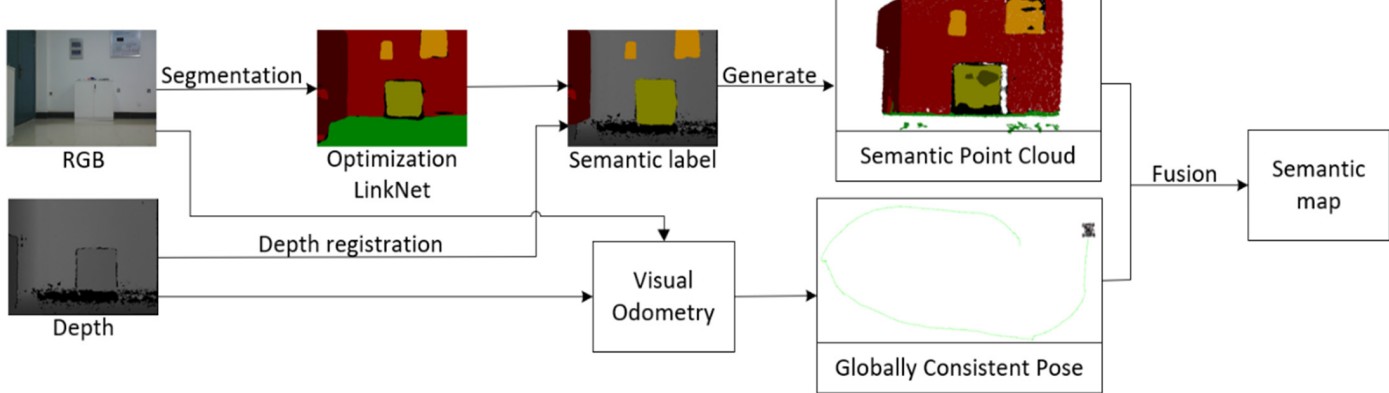

**Figure 5.** Algorithm framework of semantic map construction.

## 4. Experiment

In this section, in order to validate the effectiveness of NGLSFusion, the experiments demonstration is conducted, which is divided into three parts: dataset introduction, VO module comparison, and semantic map construction performance comparison. Each module is unfolded in the experiments using offline and online runs, and tested using currently accepted and accurate rubrics.

The offline run method is chosen to expose the TUM RGB-D dataset [61]. In order to compare the speed in front-end feature point extraction, this paper evaluates the time-consumption per frame of ORB-SLAM2 [13], ORB-SLAM3 [14], and the algorithm in this paper. On the VO module, the three algorithms are compared in terms of front end per frame time-consumption and algorithm stability, and the error between the keyframe trajectory pose and ground truth of the above three algorithms is calculated using the method of Zhang et al. [62] to evaluate the accuracy using absolute translation root mean square error (RMSE). The RMSE is defined as follows:

$$RMSE_{atrans} = \left( \frac{1}{M} \sum_{i=1}^{M} \| \Delta atrans \|^2 \right)^{\frac{1}{2}} \tag{8}$$

To check the dispersion of the error between the keyframe trajectory pose and the ground truth, standard deviation (STD) is used to evaluate the error dispersion. The STD is defined as follows:

$$STD_{\Delta p} = \left( \frac{1}{M} \sum_{j=1}^{M} \| \Delta p \|^2 \right)^{\frac{1}{2}} \tag{9}$$

where $\Delta atrans$ is the error between the pose and ground truth estimated by the algorithm after its timestamp, $\Delta p$ is the difference between the pose and ground truth averages estimated by the algorithm after its timestamp, and $M$ denotes the total number of acquired poses.

The online operation method uses ROS [20], which allows the camera to communicate directly with the master controller and other sensors through topic messaging, and provides debugging and visualization tools for user-friendly operation, and, most importantly, many of the open-source algorithms used in the algorithms of this paper are implemented on ROS, such as ORB-SLAM2 [13], OctoMap [18], and Voxblox [19].

All experimental data throughout the text were analyzed on the mobile robot, which was equipped with an Intel i7-9750H CPU @ 2.60 Hz, 16 GB RAM and NVIDIA GTX1650 4G discrete graphics card, an external HDMI display, a Songling SCOUT MINI four-wheel differential mobile chassis, and a CAN to USB interface and ROS driver to provide the robot with stable operating environment. The parameters in the subsequent algorithm will be explained during the experiment.

### 4.1. Dataset Introduction

The TUM RGB-D dataset includes static environment (handheld cameras and mobile robots), fast camera moving environment, and dynamic environment, among which the latter two environments are extremely challenging for the robustness and accuracy of the algorithm. The dataset also provides real-world robot data, including blurred, dimly lit scenes, scenes without distinctive features, and rapidly transforming scenes. In this paper, the above datasets are tested and validated to give the most real experimental results and detailed analysis. Most importantly, the dataset uses a highly accurate, time-synchronized motion capture system to record real camera poses and encapsulate them into an image sequence that provides a very accurate reference standard for this paper.

### 4.2. VO Module Comparison

The experiments in this section select sequences from eight TUM RGB-D datasets for validation, which contain the various complex environments described above. The average time-consumption of ORB-SLAM2 [13], ORB-SLAM3 [14], and the algorithm in this paper were compared by calculating the feature point extraction time-consumption per frame, in which the dataset of fr2/slam, a mobile robot viewpoint, has spatial jumps, which led to visualization jams in ORB-SLAM3, so these data were not used. The average time-consumption for feature point extraction is shown in Table 1.

**Table 1.** Comparison test of the three algorithms in terms of feature point extraction speed per frame (ten experiments are taken as the average value, in milliseconds).

| Sequence Name | ORB SLAM2 | ORB SLAM3 | Ours |
|:---:|:---:|:---:|:---:|
| fr1/desk | 18.454 | 17.177 | 9.856 |
| fr1/desk2 | 18.660 | 17.072 | 9.479 |
| fr1/room | 17.088 | 15.982 | 9.342 |
| fr2/desk | 16.555 | 15.241 | 9.854 |
| fr2/slam | 15.573 | - | 7.979 |
| fr3/office | 16.621 | 15.910 | 8.418 |
| fr3/stn | 12.653 | 11.915 | 6.923 |
| fr3/rpy | 21.366 | 19.138 | 11.508 |

The experimental results show that it is nearly twice as fast as the original ORB-SLAM2 in terms of feature point extraction per frame. ORB-SLAM2 uses ORB features for front-end feature point extraction, on the basis of which ORB-SLAM3 changes the original static matrix to feature matrix, and the speed is significantly improved. The results also show that the combination of ORB features, optical flow method, and direct method used in this paper to process image frames is far superior to ORB-SLAM2 and ORB-SLAM3 in terms of speed. The following is a comparison of the average time-consumption per frame by the three algorithms at the front end, as shown in Table 2 of the average front-end time-consumption.

**Table 2.** Comparison experiments on the processing time per frame of the three algorithms in the front end of the VO module (ten experiments to take the average value, in milliseconds).

| Sequence Name | ORB SLAM2 | ORB SLAM3 | Ours |
|:---:|:---:|:---:|:---:|
| fr1/desk | 39.211 | 33.819 | 30.904 |
| fr1/desk2 | 39.207 | 33.534 | 29.392 |
| fr1/room | 34.941 | 30.634 | 26.732 |
| fr2/desk | 34.408 | 30.125 | 26.109 |
| fr2/slam | 27.191 | - | 20.781 |
| fr3/office | 37.443 | 32.903 | 25.847 |
| fr3/stn | 25.932 | 22.289 | 19.049 |
| fr3/rpy | 40.540 | 37.073 | 30.372 |

The experimental results show that the improved feature point extraction method leads to a speedup in the overall VO front-end time consumption. In the VO module front end, feature point extraction and matching occupy a large amount of time. In this paper, the VO module is improved from feature point extraction, and although redundant calculations are generated when processing non-keyframes, in terms of the VO module as a whole, this paper provides a lightweight VO that can estimate the pose faster. The faster the speed of VO, the more frames are tracked, allowing VO to track more accurately in fast-moving or rotating datasets. Of course, this paper pays more attention to the stability of the algorithm, and counts the success rates of the three algorithms running datasets, as shown in Table 3.

**Table 3.** Three algorithm stability experiments (number of successes in ten experiments).

| Sequence Name | ORB SLAM2 | ORB SLAM3 | Ours |
|:---:|:---:|:---:|:---:|
| fr1/desk | 90% | 100% | 100% |
| fr1/desk2 | 90% | 100% | 90% |
| fr1/room | 100% | 100% | 100% |
| fr2/desk | 100% | 100% | 100% |
| fr2/slam | 20% | - | 30% |
| fr3/office | 100% | 100% | 100% |
| fr3/stn | 100% | 100% | 100% |
| fr3/rpy | 100% | 100% | 100% |

The experimental results show that the original ORB-SLAM2 algorithm is prone to run failure in datasets where the camera moves too fast and dynamic objects are present, while the ORB-SLAM3 algorithm completes the whole process by rebuilding the map when the tracking fails because of the presence of a multimap system. However, the algorithm in this paper does not need obvious texture features in the tracking process, and it can only track the gray changes in the image, which fully reflects its stability. The accuracies of the three algorithms are further compared while satisfying a rapidity and stability. The ORB-SLAM2 and ORB-SLAM3 gave some results processed by double standard deviation in their papers. Considering the influence of different devices on the algorithm results, the results given in this paper meet the requirements of the original paper after testing, as shown in the RMSE comparison of the three algorithm in Table 4.

The experimental results show that the VO module in this paper has the highest accuracy on seven datasets, and ORB-SLAM3 has the highest accuracy on only one dataset, respectively. By comparing the accuracy values, the VO accuracy in this paper is better than the original ORB-SLAM2 and can achieve similar accuracy to, or even higher accuracy than, ORB-SLAM3; the main reason is that the VO will perform a preoptimization after extracting the keyframe feature points, and then the back end will perform nonlinear optimization to achieve higher accuracy. For the Handheld Camera dataset, the VO in

this paper has the highest accuracy among the two datasets, fr1/desk and fr1/desk2, in which the camera moves faster, and this high accuracy is due to a fast frame-tracking VO module. ORB-SLAM3 has the highest accuracy on fr1/room, which is affected by the ambient illumination of this dataset, resulting in inaccurate pixel gray extracted by this algorithm. In two datasets, fr2/desk and fr3/stn, where the camera moves slowly, the VO in this paper is significantly superior to the original ORB-SLAM2 in accuracy, and is similar to ORB-SLAM3. In a home-environment-like dataset fr3/office, the camera is rotated along its environment for map construction, which is a test of the algorithm's loop closure performance. Fortunately, all three algorithms are able to perform accurate loop closure, but in this paper, VO is able to obtain a better accuracy due to the optimization of the pose using more information of keyframes. For the Robot dataset, the spatial jump amplitude of the fr2/slam dataset is relatively small, and the presence of debris on the ground causes the robot to shake, which results in transient tracking loss of the camera, and the subsequent tracking process is quite stable. This Robot dataset is only used for reference purposes in this paper due to their own data loss problems, and their accuracy is not judged. For the Dynamic Objects dataset, the fr3/rpy dataset will inevitably generate the feature points on the human body in 3D space in the form of map point while the camera is moving, and thus the camera will make misjudgments when solving the pose, resulting in the reduction of pose accuracy. The method in this paper can reduce the extraction of feature points on dynamic objects as a way to reduce misjudgment in pose estimation. In the case of a satisfactory accuracy result, this paper is also interested in the degree of dispersion of the trajectory error of this VO module, so the performance of the three algorithms in eight datasets is counted separately, as shown in Table 5 of the STD comparison of the three algorithms.

**Table 4.** RMSE of the three algorithms (median value after 10 tests, in meters).

| Sequence Name | ORB SLAM2 | ORB SLAM3 | Ours |
| :---: | :---: | :---: | :---: |
| fr1/desk | 0.017280 | 0.017063 | 0.016755 |
| fr1/desk2 | 0.026217 | 0.025845 | 0.025596 |
| fr1/room | 0.059723 | 0.055109 | 0.096578 |
| fr2/desk | 0.008506 | 0.007727 | 0.007046 |
| fr2/slam | 0.109209 | - | 0.075307 |
| fr3/office | 0.009770 | 0.009472 | 0.008647 |
| fr3/stn | 0.010764 | 0.010350 | 0.010117 |
| fr3/rpy | 1.149160 | 0.772202 | 0.645489 |

**Table 5.** STD of three algorithms (median value after 10 tests, in meters).

| Sequence Name | ORB SLAM2 | ORB SLAM3 | Ours |
| :---: | :---: | :---: | :---: |
| fr1/desk | 0.009794 | 0.010109 | 0.008273 |
| fr1/desk2 | 0.011452 | 0.011167 | 0.008016 |
| fr1/room | 0.034836 | 0.028755 | 0.022864 |
| fr2/desk | 0.003302 | 0.003041 | 0.002659 |
| fr2/slam | 0.048128 | - | 0.027429 |
| fr3/office | 0.004309 | 0.004108 | 0.003326 |
| fr3/stn | 0.003955 | 0.005600 | 0.003475 |
| fr3/rpy | 0.560861 | 0.338517 | 0.350338 |

The experimental results show that the VO module in this paper has the smallest STD between the trajectory estimates and the ground truth means in the seven datasets, reflecting the excellent precision of the module. For the Handheld Camera dataset, combined with Table 4 for analysis, the RMSE effect of the VO module on fr1/room in this paper is not ideal,

but its STD is the smallest, and it can be concluded that the camera trajectory estimation does not show substantial fluctuations compared to ground truth. This reason is mainly attributed to the use of the PnP-Ransac algorithm for preoptimization of camera pose and nonlinear optimization of the back end in this paper, which makes the camera trajectory estimation error smaller. For the Robot dataset, the use of RMSE and STD metrics is still not convincing, so the results are only for reference. For the Dynamic Objects dataset, the STD of VO in this paper is smaller, even half of the original ORB-SLAM2. Considering that the trajectory estimation of the algorithm in this paper cannot be visualized from the data perspective, the trajectory is depicted on the basis of the data in Tables 4 and 5, as shown in Figures 6 and 7 for the comparison of trajectory pose.

Among them, Figure 6a–c,g,i,j are the top view of the trajectory, and Figure 6d–f,h,k,l are the side view of the trajectory. The experimental results show that the VO module in this paper achieved the expected effect in estimating the camera pose. In Figure 6, by testing the performance of the module on six datasets and comparing the trajectories with the original ORB-SLAM2 and ORB-SLAM3, it is found that the module far exceeds the original ORB-SLAM2 in terms of estimated pose accuracy and is similar to ORB-SLAM3 in terms of accuracy, and is even better in terms of pose accuracy for some trajectories, which is closer to the ground truth. In this dataset, for the camera pose estimated by the VO module of this paper in the trajectory maps of (a, d) and (i, k), two viewpoints fit the ground truth more closely than the other two algorithms. The camera poses in (b, e), (c, f), and (j, l) in both views show a certain magnitude of trajectory jumps, while the trajectory estimated by VO in this paper is closer to the ground truth, which shows the precision of this module. The camera poses estimated in (g, h) views show substantial trajectory jumps, but the VO in this paper can track a large number of frames and obtain more information about the pose, so the camera poses estimated by VO in this paper are the best in the trajectory.

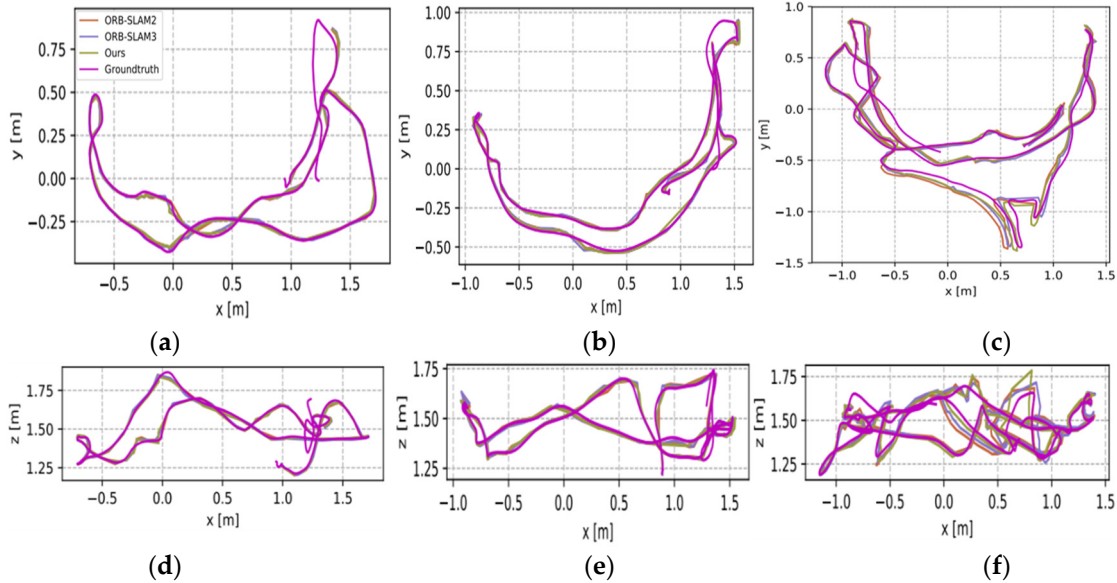

**Figure 6.** *Cont.*

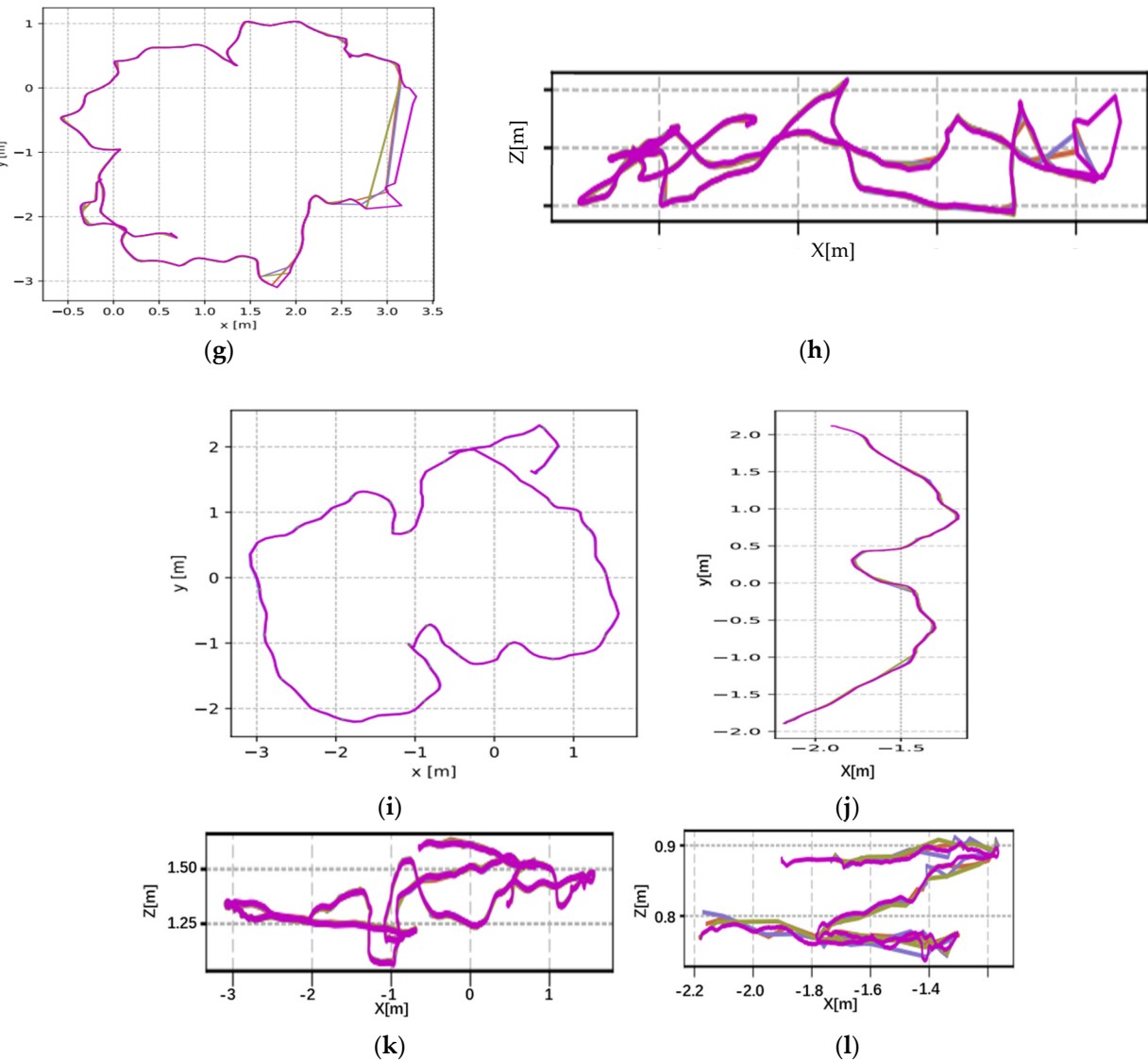

**Figure 6.** The trajectory position comparison diagram of the Handheld Camera dataset. (**a–c,g,i,j**) the top view of the trajectory; (**d–f,h,k,l**) the side view of the trajectory.

Among them, Figure 7a,c are the top view of the trajectory, and Figure 7b,d are the side view of the trajectory. In the Robot dataset, (a, b) cannot provide an accurate basis for the VO in this paper, but it can verify the robustness of the VO in this paper on the mobile robot platform theoretically and practically through RMSE, STD, and trajectory pose plots, respectively. In the Dynamic Objects dataset, the (c, d) dataset shows a serious drift when the algorithm estimates the camera pose, while the method used in this paper filters part of the dynamic points well, and thus the estimated pose is closer to the ground truth compared to the original ORB-SLAM2 and ORB-SLAM3.

In summary, this paper performs experimental validation in three parts: front-end tracking, positional estimation, and trajectory depiction, and finally gives a VO module with fast frame tracking and high robustness.

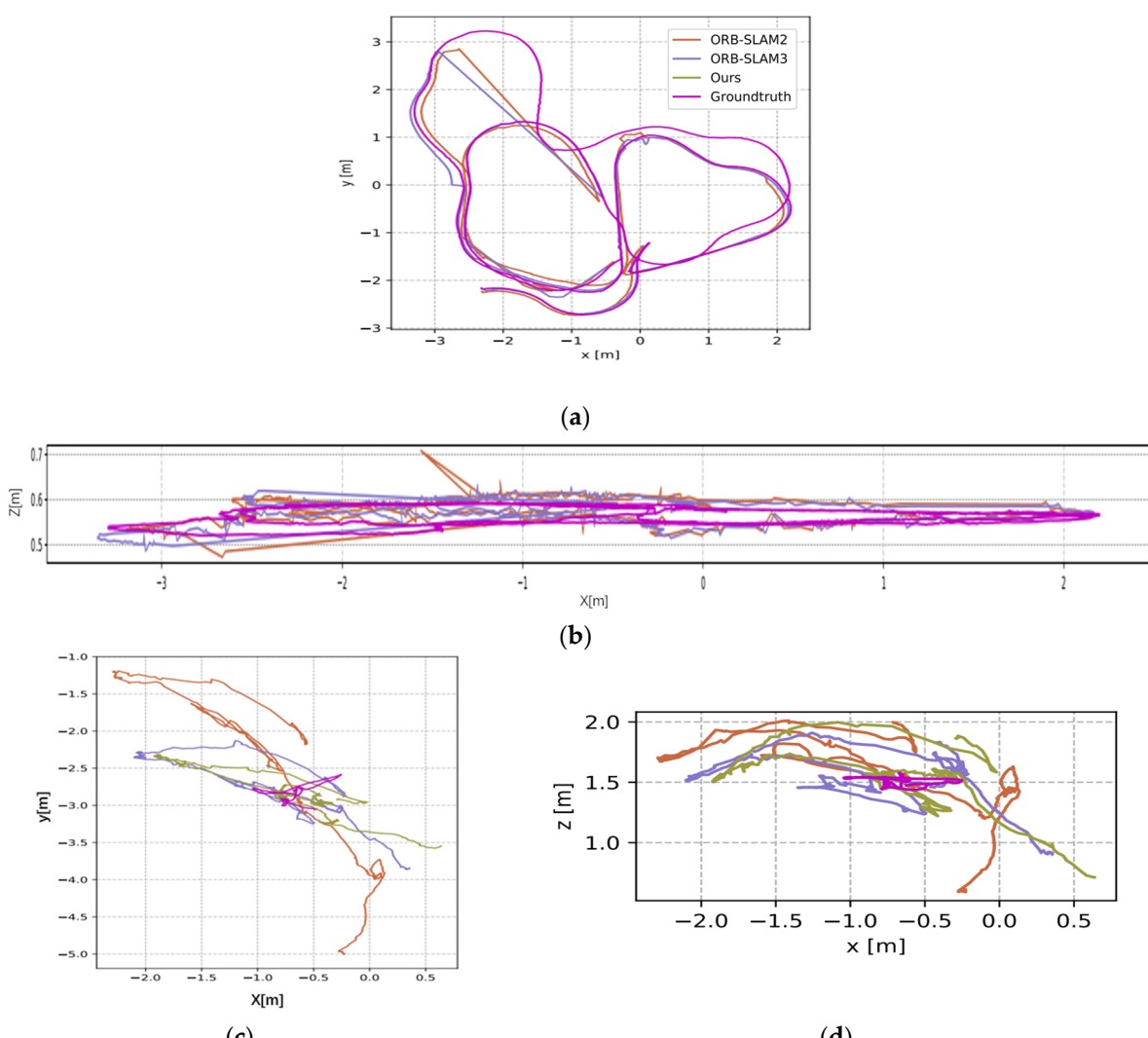

**Figure 7.** The trajectory position comparison diagram of the Robot dataset and the Dynamic Objects dataset. (**a**,**c**) are the top view of the trajectory; (**b**,**d**) the side view of the trajectory.

### 4.3. Semantic Map Construction Performance Comparison

In this section, two scenes are selected to verify the algorithm of this paper: one is a small office scene with length and width of 6.6 m × 3.3 m, respectively. The other is a large laboratory scene with length and width of 14.4 m × 6.6 m, respectively. The above two scenes' data are all run in real time on the robot with the configuration given in this section, and the image information is captured by the AstraPro camera equipped with the robot, which is trained on the public dataset SUNRGBD [63] in this paper to obtain a LinkNet pretraining model that uses different colors to differentiate the semantic information of each class of objects, as shown in Table 6, which outlines the semantic information of objects in indoor environments.

**Table 6.** Semantic information of objects in indoor environments.

| Objects | wall | floor | cabinet | door | chair | mural | desk |
|---|---|---|---|---|---|---|---|
| Color | | | | | | | |

In this paper, we conducted extensive experiments using the semantic colors and categories of objects given in Table 6, in which we found that it is difficult to restore the environment semantic map in real time if running on devices with limited hardware performance. For the semantic map construction performance, this paper will be compared with building maps using CPU and using GPU, and a review of the algorithm will be conducted in four aspects: semantic segmentation time-consumption per frame, local semantic map time-consumption per frame generated, map building effect, and GPU occupancy of the three methods, and to ensure the consistency of experimental data, the resolution of building maps will be uniformly set to 0.02 m. In the subsequent experiments, UCSSLAM (Use-CPU Semantic SLAM) and UGSSLAM (Use-GPU Semantic SLAM) are used to represent the methods of constructing semantic maps using CPU and GPU, respectively.

### 4.3.1. OctoMap

Considering that there is no professional equipment to obtain the ground truth, a color map of the two environments is given as a reference for comparison, using [64]. When deep learning is used to extract semantic information about objects in the environment, GPU is usually used to speed up the algorithm to achieve a satisfactory real-time performance. In order to give a quantitative evaluation, the frame time-consumption between the previous frame ($Lt$) ID containing semantic information and the current frame ($Ct$) ID is defined, and the semantic frame time-consumption is defined as follows:

$$d = FrameID(Ct) - FrameID(Lt) \tag{10}$$

Therefore, the first experiment is designed in this section to give a comparison of the real-time performance using CPU, GPU, and the methods in this paper, as shown in Table 7 of the semantic segmentation time per frame for the three methods.

**Table 7.** Three methods of semantic segmentation per frame time-consumption table (ten experiments to take the average value, in seconds).

| Environment | UCSSLAM | UGSSLAM | Ours |
|:---:|:---:|:---:|:---:|
| Office | 0.40 | 0.07 | 0.14 |
| Laboratory | 0.45 | 0.08 | 0.16 |

The experimental results show that with the support of powerful GPU arithmetic, UGSSLAM processing speed per frame segmentation is the fastest, almost 1/6 of the time-consumption by UCSSLAM processing per frame. The algorithm in this paper also uses CPU for processing, which is 1/3 of UCSSLAM in terms of time-consumption per frame. Although there is still a gap in real-time performance with UGSSLAM using GPU acceleration, the application in mobile robots already meets the real-time requirement. Without relying on GPU, the algorithm in this paper gives a very desirable real-time result, based on which the second experiment in this section will be centered on generating local semantic map time-consumption per frame, because the process of generating local semantic maps contains local semantic point cloud generation, local OctoMap generation, and local map with fused semantic point cloud. Table 8 shows the three methods used to generate local semantic map time-consumption per frame.

**Table 8.** Three methods to generate per frame of local semantic map time-consumption table (ten experiments to take the average value, in seconds).

| Environment | UCSSLAM | UGSSLAM | Ours |
|:---:|:---:|:---:|:---:|
| Office | 0.92 | 0.47 | 0.54 |
| Laboratory | 0.93 | 0.48 | 0.56 |

The experimental results show that the whole semantic map construction process consumes a lot of time with or without using GPU, and these times are mainly consumed in the local map process of fusing semantic point clouds, so that the robot's perception of the environment and the generation of semantic map detail textures are maximally affected. The advantage of using GPU in UGSSLAM is mainly in generating semantic point clouds using segmentation results combined with depth information. However, this advantage does not excel in mobile robots with limited hardware devices. Compared with UCSSLAM, when only one CPU is used, the speed of generating local semantic maps is improved by 41%, which is undoubtedly exciting. The above two experiments verify the feasibility of the algorithm in this paper in terms of building maps in real time, but this cannot be judged as a key factor. Two scene semantic maps constructed by the robot in real time are given below: the global semantic map of small scenes in the office of Figure 8 and the global semantic map of large scenes in the laboratory of Figure 9.

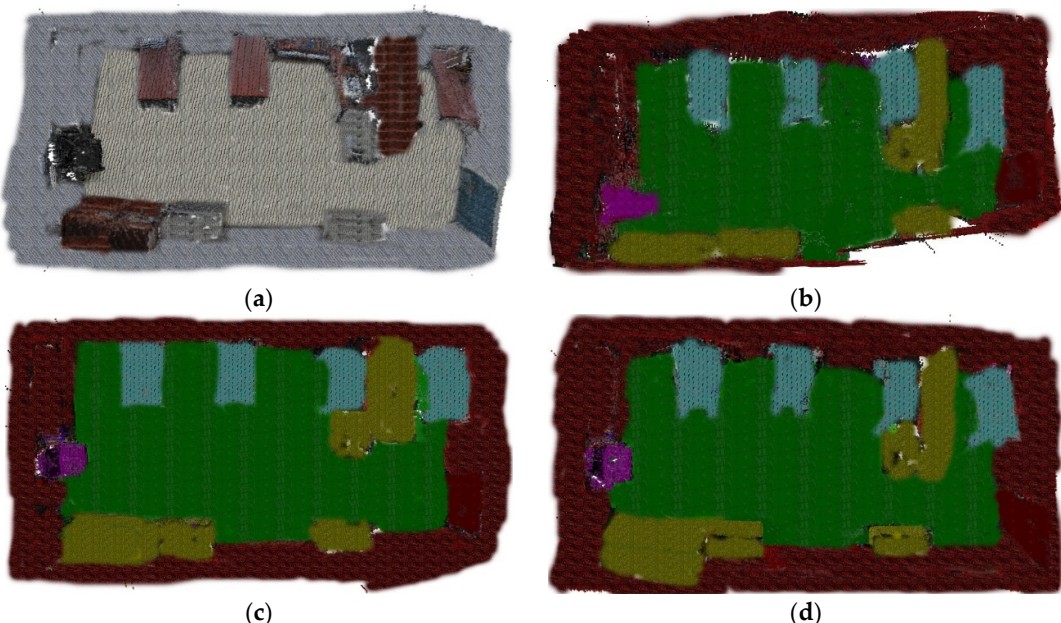

**Figure 8.** Global semantic maps for small office scenes. (**a**) Color reference map. (**b**) Semantic maps constructed by UCSSLAM. (**c**) Semantic maps constructed by UCSSLAM. (**d**) Semantic maps constructed by our method.

The experimental results show that the semantic maps constructed by the algorithm in this paper can fully restore the location of semantic objects in the environment, as in Figures 8a,d and 9a,d, and color them correctly according to the set semantic objects. Compared with the semantic map of the office constructed by UCSSLAM, the algorithm in this paper highlights its superiority in constructing details of walls, floors, and objects, where UCSSLAM cannot support accurate loop closure when the robot returns to the origin, while this paper, again relying only on CPU processing, is able to obtain a complete and closed-loop map, as shown in Figure 8b,d. The office semantic map constructed by the algorithm in this paper achieves almost similar accuracy to that constructed by UGSSLAM, all thanks to a fast-tracking frame VO module that provides continuous compact input frames for the semantic map building module in this paper; but the overall performance of UGSSLAM is better, as shown in Figure 8c,d. The semantic map construction performance in large scenes is a great challenge for almost all semantic map construction algorithms, and the semantic maps constructed by UCSSLAM show severe distortion and incorrect semantic mapping, while the algorithm in this paper can be constructed accurately and mapped correctly without GPU arithmetic support in large scenes, as shown in Figure 9b,d. In Figure 9c,d, UGSSLAM relies on the powerful arithmetic power of GPU to give high-

performance semantic mapping of objects, while the algorithm in this paper reverts to a similar extent and relies only on CPU; but in individual objects such as wall murals, the method in this paper still has incorrect semantic mapping. In this paper, semantic maps of two scene environments are given in terms of environment geometry reconstruction, object semantic mapping, and overall map building effect. The complete performance is close to the accuracy of building maps using GPU, which makes it a possibility for mobile robots to perform a complete 3D reconstruction in indoor scenes. To ensure that it can be used on devices with limited computational resources, a fourth experiment is designed in this section to count the GPU dependency of the three methods, as shown in Figure 10 (GPU occupancy).

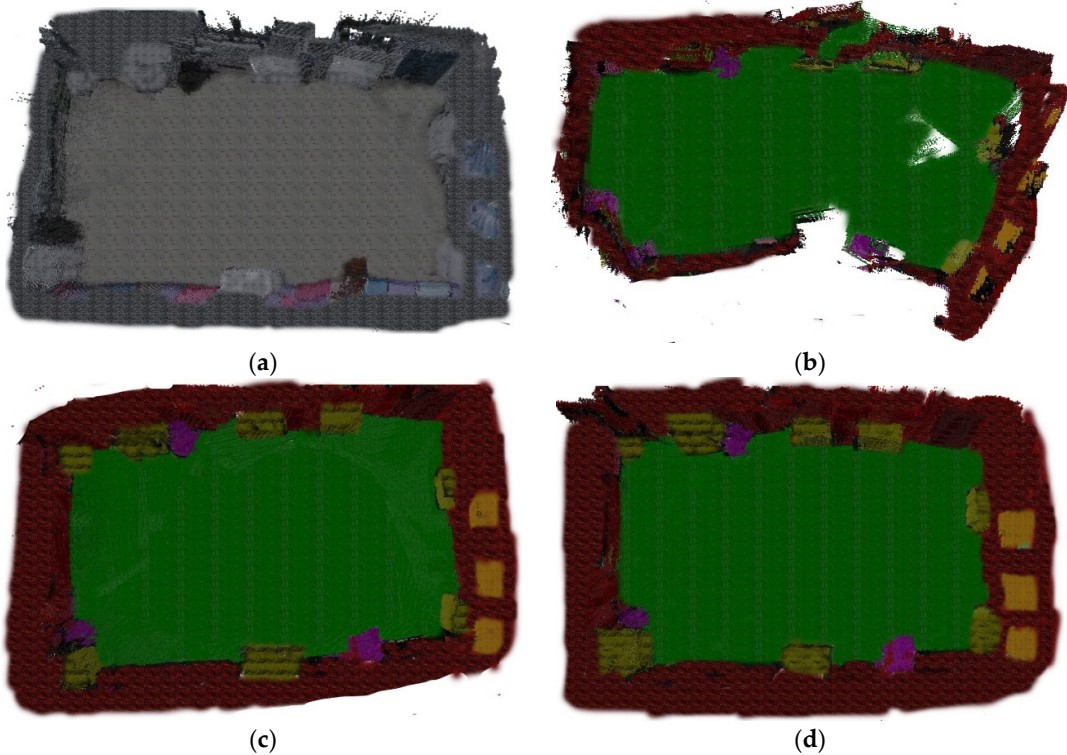

(**a**)  (**b**)

(**c**)  (**d**)

**Figure 9.** Global semantic map of large laboratory scenes. (**a**) Color reference map. (**b**) Semantic maps constructed by UCSSLAM. (**c**) Semantic maps constructed by UCSSLAM. (**d**) Semantic maps constructed by our method.

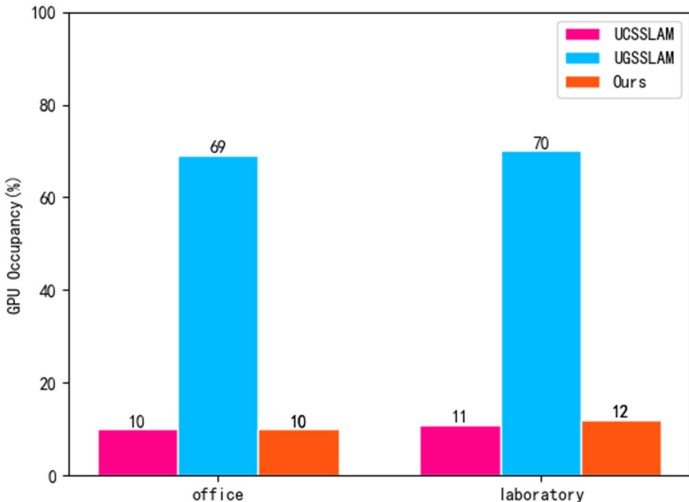

**Figure 10.** GPU occupancy.

The experimental results show that the dependence of UCSSLAM and the algorithms in this paper on GPU is much smaller than that of UGSSLAM. This undoubtedly provides a strong practical support for the algorithms in this paper to build indoor semantic map on the devices with limited computing power, where in order to ensure the uniqueness of the running environment and the validity of the experimental data, all three methods are experimented on devices configured with GPU. The VO module in tracking image frames will cause all three methods to use GPU, but the occupied GPU is small, so it is logical that both UCSSLAM and the algorithm in this paper have a small GPU occupation when building maps.

### 4.3.2. Voxblox

In the absence of an accurate ground truth, the original Voxblox [19] map was used to provide a color map of the environment, and Kimera [55] was used to provide a comparable semantic map. To avoid experimental reproducibility, only semantic map construction experiments are provided in this subsection to verify the reliability of the algorithms in this paper in terms of construction accuracy, as shown in Figures 11 and 12.

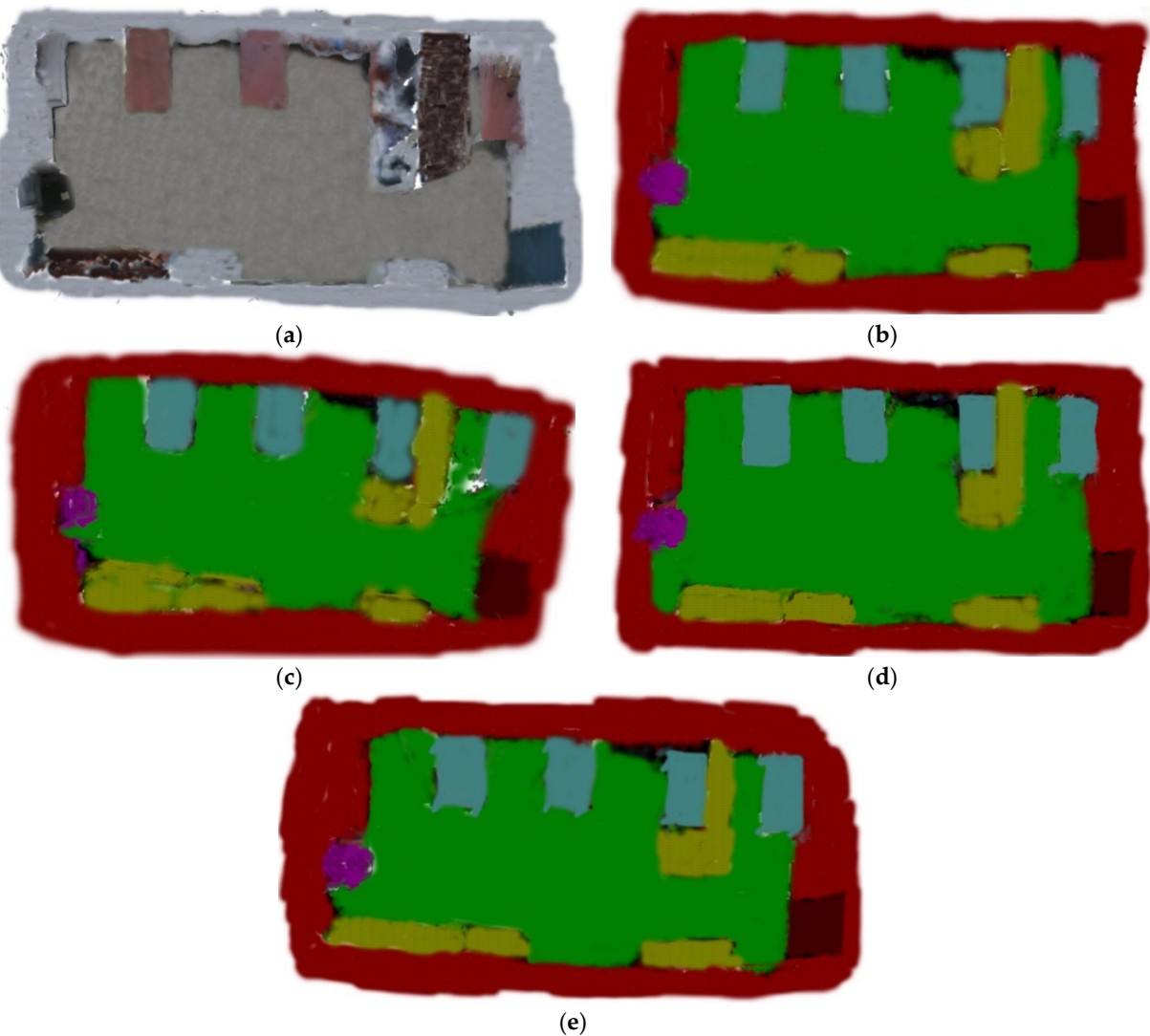

**Figure 11.** Global texture map for small office scenes. (**a**) Color reference map. (**b**) Semantic map constructed by Kimera. (**c**) Semantic maps constructed by UCSSLAM. (**d**) Semantic maps constructed by UCSSLAM. (**e**) Semantic maps constructed by our method.

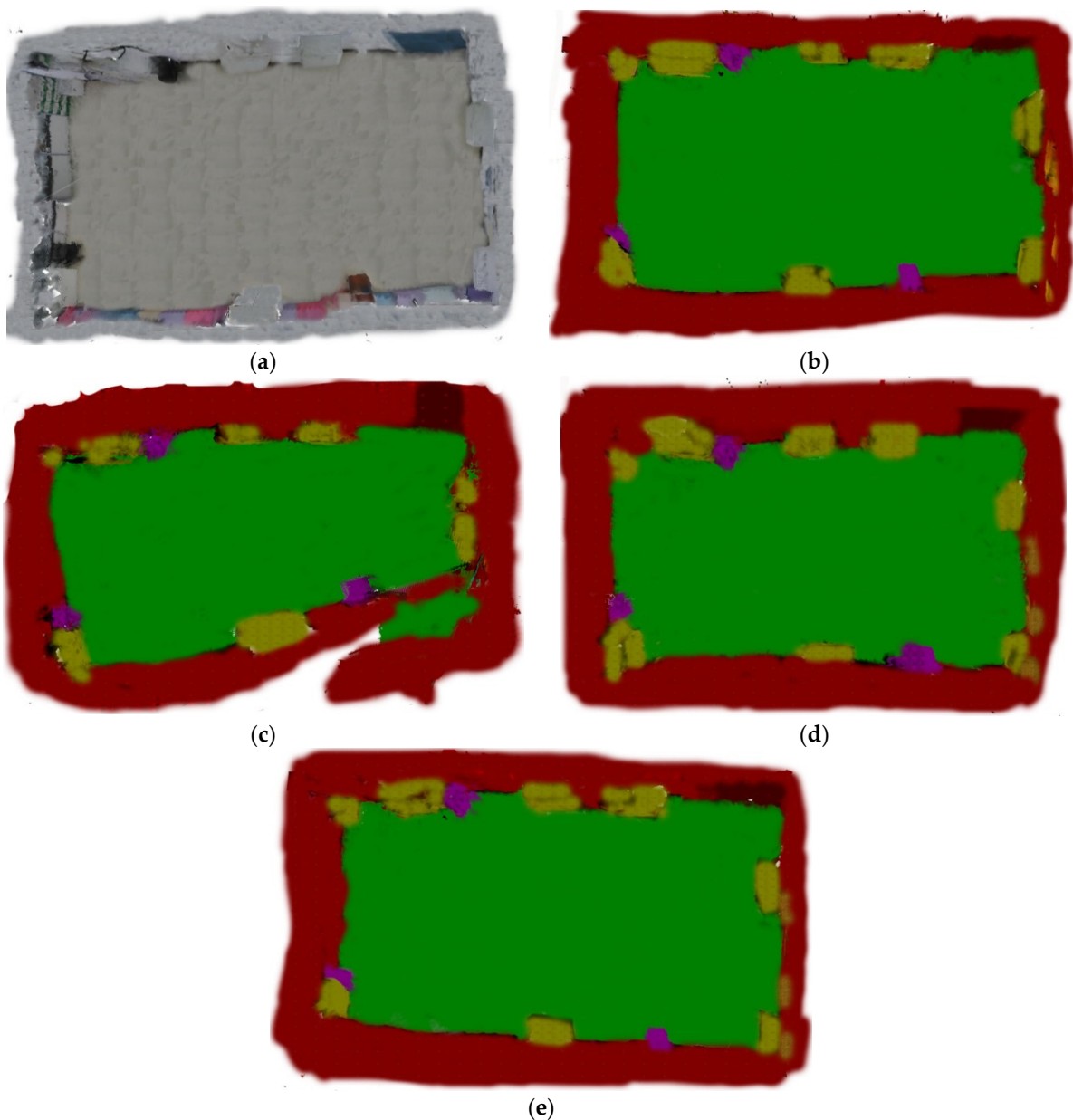

**Figure 12.** Global texture maps for large laboratory scenes. (**a**) Color reference map. (**b**) Semantic map constructed by Kimera. (**c**) Semantic map constructed by UCSSLAM. (**d**) Semantic map constructed by UCSSLAM. (**e**) Semantic map constructed by our method.

The experimental results show that the algorithm in this paper completely restores the objects in the environment in two scenes and assigns the given semantic colors to the objects, as shown in Figures 11a,e and 12a,e, and the constructed semantic texture maps can truly restore the object positions as well as shapes in the environment. Compared with the maps constructed by Kimera, the algorithm in this paper is closer to the real objects in terms of the performance of object detail restoration, but Kimera uses offline processing, and its overall performance is better, as shown in Figure 11b,e. In the case of using only CPU to build the map, the algorithm in this paper is superior to UCSSLAM in terms of object details in the environment as well as the overall map building performance. The map cabinet built by UCSSLAM shows partial distortion and partial incomplete reconstruction of the ground, as shown in Figure 11c,e. It is exciting that the algorithm in this paper has almost the same performance effect as the map constructed by UGSSLAM, which is very important for semantic SLAM without using GPU. For the semantic map construction

algorithm in large scenes, one is more concerned with whether the scene is realistically restored, and the increase of the scene also leads to the increase of the cumulative error of the algorithm, which can also test the robustness of the algorithm.

Compared with the map constructed by Kimera, the algorithm in this paper handles the details on the wall mural as well as the chair in the lower left corner better, but Kimera uses offline processing and its completeness comes through, as shown in Figure 12b,e. With regard to Figure 12c,e, the reason why UCSSLAM cannot obtain satisfactory results without the use of GPU is that UCSSLAM cannot process the key frames provided by the VO module in time. In terms of the smoothness of the constructed maps, UGSSLAM works best, but the algorithm in this paper gives similar results with only one CPU, as shown in Figure 12d,e.

## 5. Conclusions

In this paper, we propose an SLAM system for indoor semantic map construction on a mobile robot with limited computational power by combining the pose acquired by visual SLAM with the semantic information acquired by semantic segmentation in a lightweight way, named NGLSFusion. The system includes a fast frame tracking and highly robust VO module, a real-time semantic point cloud construction module, and a lightweight semantic map construction module. These three modules can provide an effective environment map information for robots in unfamiliar environments. Among them, a VO module for fast frame tracking gives accurate globally consistent poses, the robustness of which is verified with publicly available datasets. The optimized semantic segmentation pretraining model gives similar performance for semantic maps built by the robot without GPU to those built with GPU. In future research, the robot will be placed in a dynamic environment for accurate localization and mapping.

In the VO module, this paper reduces the feature point extraction time by 50% for key frames, and the front-end processing speed of VO is verified to be faster and provide an accurate pose in the subsequent experiments. In the semantic map construction module, this paper achieves a speed and accuracy close to that of GPU construction without using GPU.

However, there are still some problems in the semantic map construction method presented in this paper. While pursuing real time, the requirement of segmentation accuracy was abandoned. In the future, some other networks, such as PSPNet, will be tried as a method to improve segmentation accuracy and real-time performance, where the robot returns to the beginning, which has been ported to the robot device used in this paper and tested in large and small environments, respectively, and the performance of each detection is good.

**Author Contributions:** Conceptualization, L.W. and L.J.; methodology, L.W.; software, L.W.; validation, L.W., L.J. and Y.L.; formal analysis, B.L.; investigation, L.J.; resources, L.J.; data curation, L.J.; writing—original draft preparation, L.W.; writing—review and editing, L.J. and B.T.; visualization, L.W.; supervision, H.L.; project administration, L.J.; funding acquisition, L.J. All authors have read and agreed to the published version of the manuscript.

**Funding:** This work was partially supported by the National Key R&D Program of China grant (No.2019YFB1310000), National Natural Science Foundation of China grant (No.51874217), and Hubei Key R&D Program Project of China grant (No.2020BAB098).

**Institutional Review Board Statement:** Not applicable.

**Informed Consent Statement:** Not applicable.

**Data Availability Statement:** Computer Vision Group—Datasets-RGB-D SLAM Dataset and Benchmark (tum.de) (TUM RGB-D dataset). SUN RGB-D: A RGB-D Scene Understanding Benchmark Suite (princeton.edu) (SUNRGBD dataset).

**Conflicts of Interest:** The authors declare no conflict of interest.

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
