# Peer review of "NGLSFusion: Non-Use GPU Lightweight Indoor Semantic SLAM"

_applsci, doi:10.3390/app13095285_

Round 1
Reviewer 1 Report
The manuscript "NGLSFusion: Non-Use GPU Lightweight Indoor Semantic SLAM" is presented in a good way how ever some of issues must be solved to increase the effectiveness and readability .
1. Symbol in all the equations may be defined in the context.
2. In recent years, more attention has been focused on combining 3D reconstruction algorithms with deep learning algorithms and building semantic SLAM systems, and so far, these two research fields have been revitalized and revitalized by combining with each other. The more popular ones at this stage are dynamic. In line number 44,45 The authors used revitalized and revitalized make it simpler or use only one word as both gives same meaning.
3. Some Components in figure 1 are blur the authors are requested to modify the figure.
4. In over all manuscript Spaces must be provided between sentences and citations such as monocular vision[21-25], The application of feature point method in VO(VIO), such as Horn-Schunck optical flow method[32]), (such as Lucas-147 Kanade optical flow method [33]) etc.
5. The caption of figure 6 is too long . The details should be removed and may be added in the paragraph form.
6. Add some contribution such as your findings results in conclusion section.
A little bit shot sentences and simpler English may be used to enhance the readability of the manuscript .
Author Response
非常感谢您为审阅本文所做的热情努力。您的建议非常有意义,帮助我们实现了自己在专业问题和英语表达方面的短板,并且对论文有很大帮助。现对你的建议作出如下回应,新稿件中着重指出了更正之处。

Reviewer 2 Report
This paper proposes a new method that can construct lightweight indoor semantic maps without using GPU. The proposed method systematically integrates RGBD-based Visual Odometry (VO), semantic segmentation and semantic reconstruction in real time on a mobile robot. Experimental results show that the proposed method improves the tracking speed with accuracy of camera pose comparable with that of methods that use GPUs. The proposed method is original and may have applications in practice. However, the following issues need to be addressed before the paper can be accepted for publication.
1. The authors should clearly describe the major contributions of the proposed method in Section 1.
2. In Section 3.2, what is the threshold used to determine wrong tracking points? In general, what is the percentage of wrong tracking points in practice?
3. In line 267, page 7, there is no si in Equation (5), do you mean s instead? In Equation (7), the mapping pi needs to clearly described. Is it related to Equation (5)?
4. Which dataset was used to train the light-weight network for semantic segmentation?
5. The algorithm presented in page 9 after Figure 4 needs two sets of semantic information as its input, can the authors explain why such a comparison is necessary and how the two sets of semantic information were obtained? In addition, do you mean “optical” or “optimal”?
6. In Figures 7 and 8, the semantic maps constructed by the proposed method are compared with those constructed by UCSSLAM, can the authors provide a quantitative evaluation of the semantic segmentation accuracy of the proposed approach and compare it with that of UCSSLAM?
7. Figure 9 shows that the GPU dependency of the proposed method is higher than zero, however, it is claimed in abstract that the proposed approach does not use GPU. Can the authors provide a better explanation on this?
The paper is generally well written but there are minor grammatical mistakes that need to be corrected.
Author Response
Thank you very much for your enthusiastic efforts in reviewing the paper. Your suggestions are very meaningful, helping us to realize our own shortcoimgs in professional issues and English expressions, and have helped the paper tremendously. We now respond to your suggestions as follows, with the corrections highlighted in the new manuscript.

Reviewer 3 Report
The authors propose a Non-Use GPU for lightweight indoor semantic map construction algorithm, named NGLSFusion. The approach presented in this research has been shown to successfully create a fully structured rebuilt semantic map and maintain camera position accuracy in real-world experiments.
Overall, the paper is nicely written. I have some comments as follows.
1. Please check the caption of Figure 2.
2. What is the last sentence of literature review, "Section 3 a detailed description of this method."?
3. How do you fuse (which technique) in Figure 5?
4. What is it? "Zhang et al 62 to evaluate the accuracy"
5. Figure 9 is not clear.
6. There is no comparison with recent papers (years 2022, 2023).
A moderate English editing is required.
Author Response

(The authors gave the same response as above.)

Reviewer 4 Report
In this manuscript, an NGLSFusion algorithm is proposed for lightweight indoor semantic map construction. The proposed approach combines RGBD-based visual odometry, semantic segmentation, and real-time semantic reconstruction on a mobile robot. Experiments were conducted on publicly available datasets to verify the effectiveness of the proposed approach, and the performance was evaluated using relevant metrics. The results are presented and discussed. Furthermore, the proposed approach is compared with other recent relevant approaches and the obtained results are discussed. In addition, the limitations of the presented approach and future directions of work are listed.
The authors have presented their work nicely, but it needs polishing. The paper is technically sound, and the references provided by the authors are applicable and relevant (64 citations).
Please consider the following corrections and comments:
#) I suggest the authors that rephrase abstract, e.g., the awkward sentence "The algorithm in this paper is the first systematically to integrate RGBD-based Visual Odometry (VO), semantic segmentation and semantic reconstruction in realtime on a mobile robot." So, the abstract needs to be improved, also the writing style should be reconsidered.
#) Regarding the sentence (Line 212): “Section 4 gives the experimental results are to verify the feasibility of NGLSFusion. “The sentence at the end of the paragraph does not fit the context. I suggest that the authors rephrase this sentence or remove it from the paragraph.
#) Regarding figures: make sure that the entire figure is on one page. Nevertheless, also make sure that the caption of the figure and the figure are on the same page.
##) Figure 3 is divided into two pages, the entire figure should be on one page.
##) Figure 6 is divided into two pages, the entire figure should be on one page.
##) Figure 8, Figure 9, Figure 11: The caption and the Figure should be on the same page.
##) Please correct the figure caption for Figure 2. “This is a figure. Schemes follow the same formatting. “. It looks like the text from the template was accidentally left in place.
#) The references provided by the authors are relevant and sufficient (64 citations). References should include the DOI according to the journal template.
#) Please check the proofreading and English spelling. There are some typos such as "equip-ped". Also, the line spacing, which should be the same throughout the text, needs to be revised.
#) Please check the proofreading and English spelling. There are some typos such as "equip-ped".
Author Response

(The authors gave the same response as above.)

Round 2
Reviewer 2 Report
All issues have been carefully addressed. I have no other concerns and would like to recommend the acceptance of the paper.
Author Response
Dear reviewing teachers and editing teachers,
Thank you to several teachers for reviewing my manuscript in their busy schedule and putting forward their valuable comments. According to the suggestions, we have revised and explained one by one. We itemize the answers to match each reviewer’s original comment.
Thank you again for the time you and the reviewers took to provide us with a better manuscript. If you have any other questions, please feel free to let us know.
Best regards,
Faithfully yours,
Le Wan

Reviewer 4 Report
After the first revision, the authors uploaded their point-by-point response to the reviewers' comments, and the authors made the changes. My concerns were mostly addressed, although corrections to the captions should still be made in the current version of the manuscript.
Please consider the following corrections and comments:
#) Regarding figures: make sure that the caption of the figure and the figure are on the same page.
##) Figure 6: Please correct the figure caption according to the journal template “If there are multiple panels, they should be listed as: (a) Description of what is contained in the first panel; (b) Description of what is contained in the second panel.”
##) Figure 7: The caption and the Figure should be on the same page.
Author Response

(The authors gave the same response as above.)
